# Recurrent WNT pathway alterations are frequent in relapsed small cell lung cancer

Alex H. Wagner [1], Siddhartha Devarakonda[2,3], Zachary L. Skidmore[1], Kilannin Krysiak [1,2], Avinash Ramu [1], Lee Trani[1], Jason Kunisaki[1], Ashiq Masood[2,3,9], Saiama N. Waqar[2,3], Nicholas C. Spies[1], Daniel Morgensztern[2,3], Jason Waligorski[1], Jennifer Ponce[1], Robert S. Fulton[1], Leonard B. Maggi Jr.[2,3,4], Jason D. Weber[2,3,4], Mark A. Watson[3], Christopher J. O'Conor[5], Jon H. Ritter[5], Rachelle R. Olsen[6], Haixia Cheng[6], Anandaroop Mukhopadhyay[6], Ismail Can [6], Melissa H. Cessna[7], Trudy G. Oliver[6], Elaine R. Mardis[1,3,8,10], Richard K. Wilson[1,3,8,10], Malachi Griffith[1,2,3,8], Obi L. Griffith [1,2,3,8] & Ramaswamy Govindan[2,3]

Nearly all patients with small cell lung cancer (SCLC) eventually relapse with chemoresistant disease. The molecular mechanisms driving chemoresistance in SCLC remain uncharacterized. Here, we describe whole-exome sequencing of paired SCLC tumor samples procured at diagnosis and relapse from 12 patients, and unpaired relapse samples from 18 additional patients. Multiple somatic copy number alterations, including gains in *ABCC1* and deletions in *MYCL, MSH2,* and *MSH6*, are identifiable in relapsed samples. Relapse samples also exhibit recurrent mutations and loss of heterozygosity in regulators of WNT signaling, including *CHD8* and *APC*. Analysis of RNA-sequencing data shows enrichment for an ASCL1-low expression subtype and WNT activation in relapse samples. Activation of WNT signaling in chemosensitive human SCLC cell lines through APC knockdown induces chemoresistance. Additionally, in vitro-derived chemoresistant cell lines demonstrate increased WNT activity. Overall, our results suggest WNT signaling activation as a mechanism of chemoresistance in relapsed SCLC.

[1] McDonnell Genome Institute, Washington University School of Medicine, St. Louis, MO 63108, USA. [2] Division of Oncology, Department of Medicine, Washington University School of Medicine, St. Louis, MO 63110, USA. [3] Alvin J Siteman Cancer Center, Washington University, St. Louis, MO 63110, USA. [4] ICCE Institute, Washington University School of Medicine, St. Louis, MO 63110, USA. [5] Department of Pathology and Immunology, Washington University School of Medicine, St. Louis, MO 63110, USA. [6] Department of Oncological Sciences, University of Utah, Huntsman Cancer Institute, Salt Lake City, UT 84112, USA. [7] Intermountain Healthcare BioRepository and Department of Pathology, Intermountain Healthcare, Salt Lake City, UT 84103, USA. [8] Department of Genetics, Washington University School of Medicine, St. Louis, MO 63110, USA. [9] Present address: Saint Luke's Health System, Kansas City, MO, USA. [10] Present address: Nationwide Children's Hospital, Columbus, OH, USA. These authors contributed equally: Alex H. Wagner, Siddhartha Devarakonda. Correspondence and requests for materials should be addressed to M.G. (email: mgriffit@wustl.edu) or to O.L.G. (email: obigriffith@wustl.edu) or to R.G. (email: rgovindan@wustl.edu)

Lung cancer is the leading cause of cancer-related death. Nearly 13% of patients with lung cancer are diagnosed with small cell lung cancer (SCLC)[1]. SCLC is a highly aggressive malignancy that is characterized by a short doubling time and tendency to metastasize early. Although patients with newly diagnosed SCLC often achieve dramatic responses to initial treatment with platinum doublet chemotherapy, most patients relapse with rapidly worsening disease that is resistant to further treatment. The median survival in patients with SCLC is ~7 months, and this has not substantially changed in the past few decades—owing to the lack of effective treatment options for patients with relapsed disease[2]. Therefore, understanding the molecular basis of treatment resistance and identifying therapeutic vulnerabilities is necessary to improve outcomes in patients with relapsed SCLC.

SCLC is characterized by a variety of genomic alterations and relatively higher mutation rate per megabase (Mb) compared to many other malignancies[3,4], and is associated with heavy tobacco smoking. Comprehensive genomic analyses of human SCLC have shown inactivation of the tumor suppressors *TP53* and *RB1* to be a characteristic feature of this disease[3]. SCLC is a high grade neuroendocrine tumor, and differentiation in SCLC is driven predominantly by the lineage factors ASCL1 and NEUROD1, which play a crucial role in cell survival[5,6]. Although subtyping SCLCs based on the lineage survival factor they overexpress is not performed in the clinical setting, ASCL1 and NEURDO1 predominant SCLCs are characterized by significant differences in their genetic and epigenetic profiles. In addition to these alterations, SCLCs also show alterations in pathways that regulate neuroendocrine differentiation (such as NOTCH signaling) and amplifications in oncogenes such as *SOX2*, *MYC*, *MYCL*, and *MYCN*. While the genomic landscape of treatment-naive SCLC has been well described by different groups, very few samples profiled to date have been obtained from patients with relapsed disease in these studies[3,7–9]. Therefore, the genomic landscape of relapsed SCLC and the molecular mechanisms that drive chemotherapy resistance in this disease continue to remain poorly defined.

The primary objective of our study is to understand the molecular mechanisms underlying chemotherapy resistance in patients with relapsed SCLC, as a means to circumvent traditional chemotherapy resistance with novel therapeutic approaches. To this end, we performed whole-exome sequencing (WES) of SCLC tumor samples obtained from 30 patients following relapse (relapse samples). Representative slides from these samples were reviewed and confirmed to be SCLC by two pathologists independently (Supplementary tables 1 and 2, Supplementary Data). Corresponding chemotherapy-naive tumor samples were available for 12 patients (treatment-naive samples). Additionally, we performed tumor RNA-sequencing of samples obtained at relapse from 18 of the 30 patients. Here we highlight the key similarities and differences in the genomic landscape of primary and relapsed SCLCs. We also demonstrate how integrative analysis of RNA and DNA sequencing data from these tumors identify activation of canonical WNT signaling as a likely mechanism of chemotherapy resistance in SCLCs.

## Results

**The mutational landscape of relapsed SCLCs.** Consistent with previously published studies, the total tumor mutation burden (TMB) of single-nucleotide variants (SNVs) and small insertions and deletions (indels) was high for both chemotherapy-naive and relapsed samples, due to the strong association between heavy cigarette smoking and SCLC tumorigenesis[3,4,8,9]. TMB was not significantly different between treatment-naive and relapsed SCLC samples (8.31/Mb [range: 1.50–16.87] vs. 8.37/Mb [range: 1.38–14.04]; $p = 0.93$, unpaired $t$- test). As expected, a single tumor sample obtained from a never-smoker (SCLC17), demonstrated a much lower mutational burden (1.40/Mb in chemotherapy-naive and 1.50/Mb in relapsed samples, respectively). Tumor samples obtained from all patients with a history of smoking were enriched for $C > A \| G > T$ transversions, and the samples from SCLC17 contained a lower proportion of $C > A$ transversions. A comparison of the mutation patterns between relapse and treatment-naive samples by deconstructSigs[10] failed to show an enrichment for the pancreatic platinum-response-associated mutation signature in relapse samples (signature 3, Supplementary fig. 1), possibly owing to the similarity in context between smoking and platinum-associated mutation signatures[11], and to the overwhelming proportion of smoking-associated mutations across our samples. Consistent with previous observations made from genomes of irradiated tumors, relapse SCLC samples obtained from patients with limited stage SCLC who were treated with radiation ($N = 14$) were enriched for indels when compared to samples that were not irradiated ($N = 16$; $p = 3.67e{-}13$, unpaired $t$- test)[12]. For two individuals, metastatic sites were sectioned and sequenced, indicating little spatial heterogeneity in mutations within a metastatic site; however, distinct subsets of mutations were identified in samples collected from different metastatic sites within the same patient (Supplementary fig. 2).

We observed *TP53* and *RB1* mutations in the majority of treatment-naive (*TP53* in 11 of 12, *RB1* in 10 of 12 patients) and relapse samples (*TP53* in 29 of 30, *RB1* in 21 of 30 patients). Loss of heterozygosity (LoH) of *TP53* was observed in the one sample lacking a detectable *TP53* mutation, and *RB1* LoH was observed in seven of nine samples where no *RB1* mutations were detected. These results are concordant with previous findings that RB1 is likely altered by intronic, epigenetic or large structural rearrangements that are not identifiable by WES in the remaining two treatment-naive samples. Together, 100% and 93% of relapse SCLC samples showed a mutation or LoH in TP53 and RB1 respectively, indicating that alterations in these genes are a characteristic feature of relapsed disease, similar to prior observations of their prevalence in primary SCLCs[3]. Including *TP53* and *RB1*, a total of 30 genes were mutated at a statistically significant level using MuSiC[13] (FDR CT, $Q < 0.1$) when evaluating only relapse SCLC samples (Fig. 1)[13,14]. *COL11A1* ($Q = 8.06e{-}07$, Supplementary table 3), which codes for the alpha I chain of type XI collagen, was the most significantly mutated gene (SMG) after *TP53* and *RB1*. COL11A1 is an important component of the extracellular matrix and its dysregulation has been shown to mediate resistance to platinum chemotherapy[15].

We also observed multiple recurrent somatic copy number alterations in relapsed SCLCs by GISTIC analysis ($Q < 0.1$) (Supplementary table 4)[16], some of which are likely to play a crucial role in mediating chemotherapy resistance. Among these was *ABCC1* ($N = 7$), also known as *MRP1* (multidrug resistance-associated protein 1), an ATP-binding cassette membrane protein that transports physiologic substrates and drugs out of the cytoplasm[17]. *ABCC1* mRNA was first isolated from a multi-drug resistant SCLC cell line (H69-AR), where it is upregulated through amplification[18]. We also observed deletions in mismatch repair (MMR) genes *MSH2* ($N = 5$) and *MSH6* ($N = 5$) at a significant level in relapse samples. Treatment-naive samples in our analysis did not show either of these copy number alterations at a significant level. Additionally, *MSH6* mutations were observed only in relapse SCLC samples ($N = 3$), and none of the treatment-naive samples in our cohort. These observations

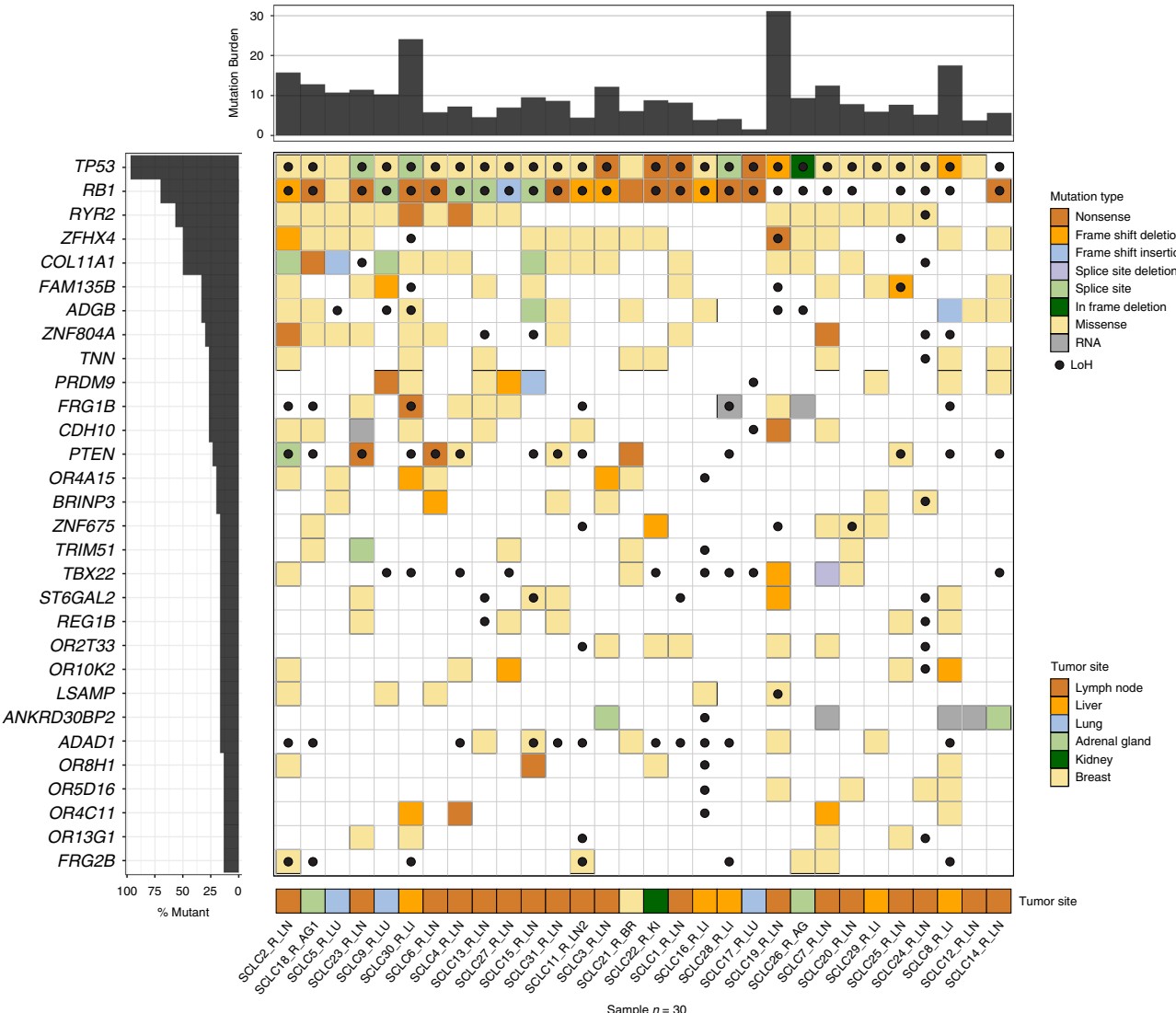

**Fig. 1** Significantly mutated genes in relapsed SCLC. Mutation burdens (mutations/mega-base pair) are displayed on the x-axis panel (top), and cohort mutation percentage by gene is displayed on the y-axis panel (left). Displayed genes are significantly mutated in relapsed SCLC compared to background mutation rates (FDR < 0.1). Each column represents an individual sample (listed at bottom). Coloring of patient-gene intersection grid denotes mutation type (top legend, center panel) and tumor site (bottom legend, bottom panel). Black dots on sample-gene squares indicate loss of heterozygosity (LoH) of that gene in that sample. TN treatment-naive, R relapsed, LN lymph node, AG adrenal gland, LU lung, BR breast, KI kidney, LI liver

are concordant with previously reported associations between MMR deficiency and cisplatin resistance in ovarian and colon cancer cell lines[19].

In addition to our relapsed mutational landscape analysis, we also compared non-synonymous mutation data between patient matched treatment-naive and relapsed sample pairs to identify relapse-acquired mutations. The high background mutation rate of SCLC and limited number of patient-matched paired samples, however, restricted our ability to statistically identify relapse-acquired mutations of interest. We therefore adopted a knowledge-based approach to identify genes of interest. We observed a total of 938 relapse-acquired mutations in 882 genes across the twelve paired samples. From these 882 genes, 603 were altered in relapse samples and none of the treatment-naive samples (Supplementary table 5). We then stringently filtered our list of 603 genes to only include genes with a known or predicted role in cancer. For this purpose, we utilized a list of 114 genes that were mutated at a statistically significant level across 21 cancer

types in The Cancer Genome Atlas (TCGA) pan-cancer analysis (Supplementary table 6)[20]. Our filtering resulted in the nomination of seven genes—*CHD8, APC, DDX3X, MBD1, RHOA, RUNX1,* and *SMARCA4* as candidates for further analysis. Among these, recurrent mutations were observed in *CHD8* (N = 6 of 30 patients), *APC* (N = 4), *MBD1* (N = 3), and *DDX3X* (N = 2).

**Chemotherapy resistance and non-neuroendocrine differentiation.** Due to the striking similarity of the mutational landscape of primary and relapsed SCLCs, we investigated the role of transcriptional changes that may account for acquired chemotherapy resistance. SCLC is a neuroendocrine malignancy with demonstrated dysregulation of neuroendocrine lineage survival factors such as ASCL1 and NEUROD1[5,6]. Previous studies of treatment-naive SCLC samples indicated that the majority of these tumors belong to a subtype that overexpresses ASCL1. ASCL1-high cell lines and murine tumors are typically enriched

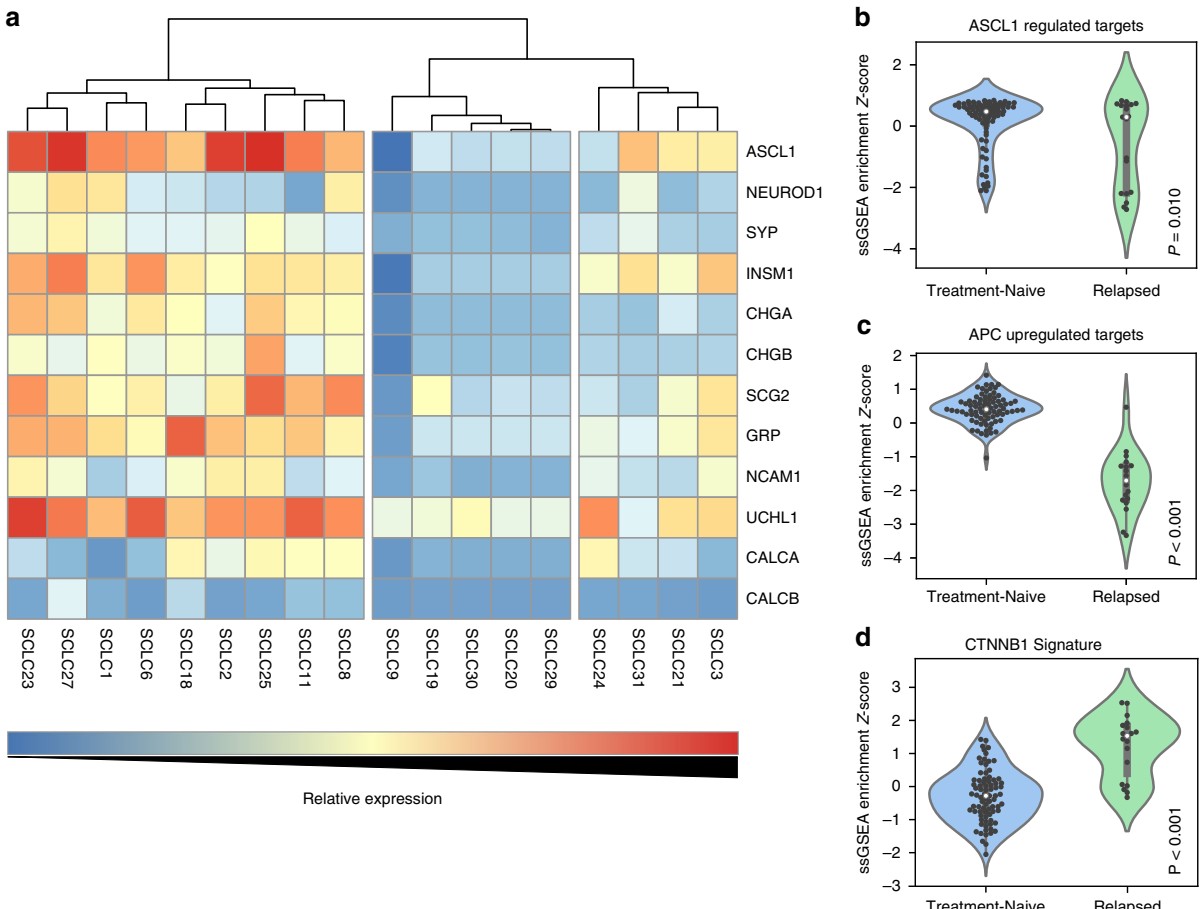

**Fig. 2** Relapsed human SCLC samples are enriched for ASCL1-low subtype and show a higher level of WNT activity. **a** Categorization of relapsed SCLC into ASCL1-high (left), NEUROD1-high (right), and ASCL1 and NEUROD1-low (dual negative; center) expression sub-types. Overall, our cohort of relapsed samples was enriched for ASCL1-low samples, when compared to treatment-naive samples that were sequenced in other studies (50% vs. 18%; $p = 0.01$, Fisher's exact test). Violin plots demonstrating differential ssGSEA enrichment scores between relapsed samples and treatment-naive SCLC samples (George et al.) for **b** ASCL1 (Borromeo et al. signature) and WNT pathway activation using **c** APC upregulated targets from Lin et al. and **d** CTNNB1 oncogenic signature from Bild et al. Gray boxes represent quartiles encompassing the mean (white dot), and whiskers extend to all data points within 1.5 interquartile range (IQR). Individual data points are overlaid as black dots. Unpaired Student's $t$ test was used to calculate two-tailed $p$ values. Color scale for **a** increases from blue (low relative expression) to red (high relative expression)

for *MYCL* amplifications or overexpression[5,6]. Contrary to what has been observed in treatment-naive SCLCs, we observed deletions encompassing *MYCL* ($N = 4$) at a statistically significant level in relapsed SCLC samples ($Q < 0.1$; Supplementary table 4). Additionally, RNA-sequencing data showed a higher than expected proportion[6] of ASCL1-low tumors (50% vs. 18%; $p = 0.01$, Fisher's exact test) in our cohort of relapsed SCLC samples (Fig. 2a).

We utilized previously published RNA-sequencing data from 80 treatment-naive SCLC samples[3] to compare their expression profiles to our relapse SCLC samples using the single-sample gene set enrichment analysis (ssGSEA) method[21]. Acknowledging the caveat that any observed differences could be due to a combination of true biological differences and batch effects across studies, we compared ASCL1-driven gene expression (defined by Borromeo et al.[5]) enrichment scores between the 80 treatment-naive and 18 relapsed SCLC samples. This showed significantly lower ssGSEA scores for ASCL1-driven gene expression among relapse samples ($p < 0.05$, unpaired $t$- test) (Fig. 2b).

To test this at the protein level, we obtained an independent set of 12 pre-treatment and 10 post-treatment human SCLC samples

from patients who received standard platinum chemotherapy. Immunohistochemistry for ASCL1 revealed a significant reduction in ASCL1-positive (ASCL1+) samples post-treatment (from 6 of 12 ASCL1+ pre-treatment to 0 of 10 ASCL1+ post treatment, $p = 0.0152$, Fisher's exact test; Fig. 3a, b). Consistent with these data, both *MYCL* and *ASCL1* were downregulated in two independent matched pairs of chemotherapy sensitive and resistant human SCLC cell lines (Fig. 3c). Collectively, these findings suggest a link between non-neuroendocrine differentiation and chemotherapy resistance in SCLCs. This hypothesis is concordant with in vitro data from other studies that have shown an association between non-neuroendocrine differentiation and chemotherapy resistance in SCLC[22,23].

**WNT signaling and chemotherapy resistance in SCLC.** Based on our observations that markers of non-neuroendocrine differentiation are linked to chemotherapy resistance, we hypothesized that chemotherapy resistance would be driven by one or more modulators of cellular differentiation. Evaluating our previously identified relapse-acquired genes, we observed that mutations in *APC* and *CHD8* were both mutually exclusive and present in

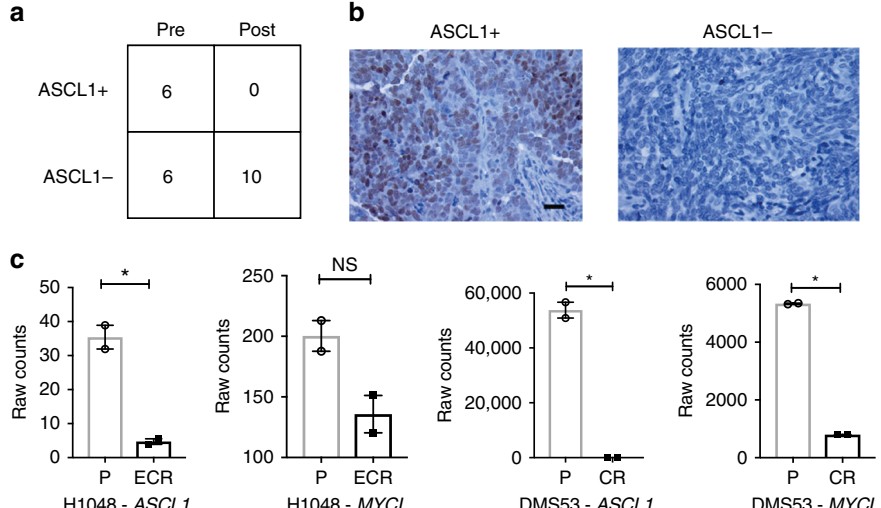

**Fig. 3** ASCL1 and MYCL are downregulated in post-chemotherapy human SCLC tissue and chemotherapy resistant cell lines. **a** Contingency table with number of chemotherapy naive (pre) and post-chemotherapy (post) SCLC human samples stained with antibodies to ASCL1. **b** Representative IHC for ASCL1 positive staining (brown) and negative staining (blue). Scale bar represents 20 μm. **c** RNA expression (counts) for indicated genes (ASCL1, MYCL) from matched pairs of chemotherapy-naive and resistant human SCLC cell lines performed in biological duplicate and compared using unpaired t- tests. H1048 NCI-H1048, P parental cells, CR cisplatin resistant, ECR etoposide and cisplatin resistant, *p < 0.05, NS not significant. Error bars are mean ± standard error

nearly one third of relapse SCLC samples (N = 4, 6, respectively; Supplementary fig. 3). *APC*, which is mutated in several cancers, is a well-established negative regulator of CTNNB1 (β-catenin) and WNT signaling[24]. Similarly, CHD8 inhibits CTNNB1-mediated transcription in the nucleus and is recurrently mutated in prostate and gastric cancers[25–28]. Due to the role of WNT signaling in cellular differentiation and in driving treatment resistance in other cancers, we hypothesized that this pathway may play an important role in the acquisition of chemotherapy resistance in relapsed SCLC[29–33]. To evaluate this, we examined our data for mutations in other genes of the CTNNB1-mediated (canonical) WNT signaling pathway (Online Methods; Supplementary table 7). This analysis revealed relapse-acquired mutations in additional genes regulating canonical WNT signaling such as *CSNK1A1*, *HECW1*, *KMT2D*, *TLE2*, *TLE3*, *WNT1*, and *YWHAZ* (Supplementary fig. 3). Overall, relapse SCLC samples from 24 of 30 patients (80%) contained non-synonymous mutations in genes belonging to canonical WNT signaling that were unaltered in treatment-naive samples, with 6 of 12 (50%) paired samples exhibiting at least one relapse-acquired non-synonymous mutation in this pathway. We also observed highly recurrent LoH in the WNT signaling genes *APC* (N = 19; 63% of cohort), *CSNK1A1* (N = 20; 67%), and *PSMD6* (N = 25; 83%). Based upon these observations, we hypothesized that the level of canonical WNT pathway activation would be significantly higher among relapsed SCLC samples when compared to treatment-naive samples. We found that WNT pathway activation (as defined by gene expression signatures from two independent studies[34,35]) showed significantly higher ssGSEA enrichment scores for canonical WNT pathway activity in relapse SCLC samples compared to primary, treatment-naive samples (p < 0.0001 for each signature, unpaired t- test; Fig. 2c, d).

We then performed in vitro analyses to test the hypothesis that WNT activation drives chemotherapy resistance in SCLC by using APC knockdown as a model for WNT activation. Stable APC knockdown through lentiviral delivery of short hairpin RNAs (shRNAs) in the NCI-H1694 cell line (chemotherapy sensitive with wildtype *APC*), led to the activation of canonical WNT signaling and made these cells highly resistant to etoposide (Fig. 4a–c) and—to a lesser degree—cisplatin (Supplementary fig. 4a, b). Additionally, these cells could be re-sensitized to chemotherapy following overexpression of a wildtype APC construct lacking untranslated regions (UTRs), in cells stably transfected with a knockdown shRNA construct targeting the 3′ UTR (construct shAPC#2; Fig. 4c and Supplementary fig. 4c). We were also able to induce etoposide resistance and activation of WNT signaling through CRISPR–Cas9-guided deletion of *APC* in a different SCLC cell line, H82 (NCI-H82) (Fig. 4d–f). The extent to which these cells became etoposide resistant was modest in comparison to H1694 cells with APC knockdown, with a 4.3-fold increase in IC50 (Fig. 4f). Next, to determine the role of *APC* and WNT signaling in the setting of acquired chemotherapy resistance in vitro, we measured WNT activity in human SCLC cell lines DMS53 and NCI-H1048 and their chemotherapy resistant variants derived in culture. Chemotherapy resistant variants of both cell lines showed evidence of increased WNT activity when compared to their chemotherapy sensitive counterparts based on increased levels of CTNNB1 and TCF/LEF (TOPFlash) WNT reporter activity (Supplementary fig. 5). Furthermore, ingenuity pathway analysis (IPA) of global transcriptomic changes in these matched chemotherapy-sensitive and resistant pairs identified CTNNB1 as a top predicted upstream regulator in resistant samples (p = 2.35e−23 and 2.45e−18 for DMS53 and NCI-H1048, respectively, right-tailed Fisher's exact test).

## Discussion

Results from our study suggest that activation of canonical WNT signaling leads to chemotherapy resistance in SCLC—a finding that has not been previously reported to the best of our knowledge. WNT activation is likely to drive treatment resistance through its effects on apoptosis, DNA repair, expression of drug efflux pumps, cell differentiation, and/or by facilitating immune evasion[30,31,36,37]. We noted alterations in multiple genes that participate in WNT signaling among relapsed tumors, independent of the presence of *APC* or *CHD8* alterations. Similar findings

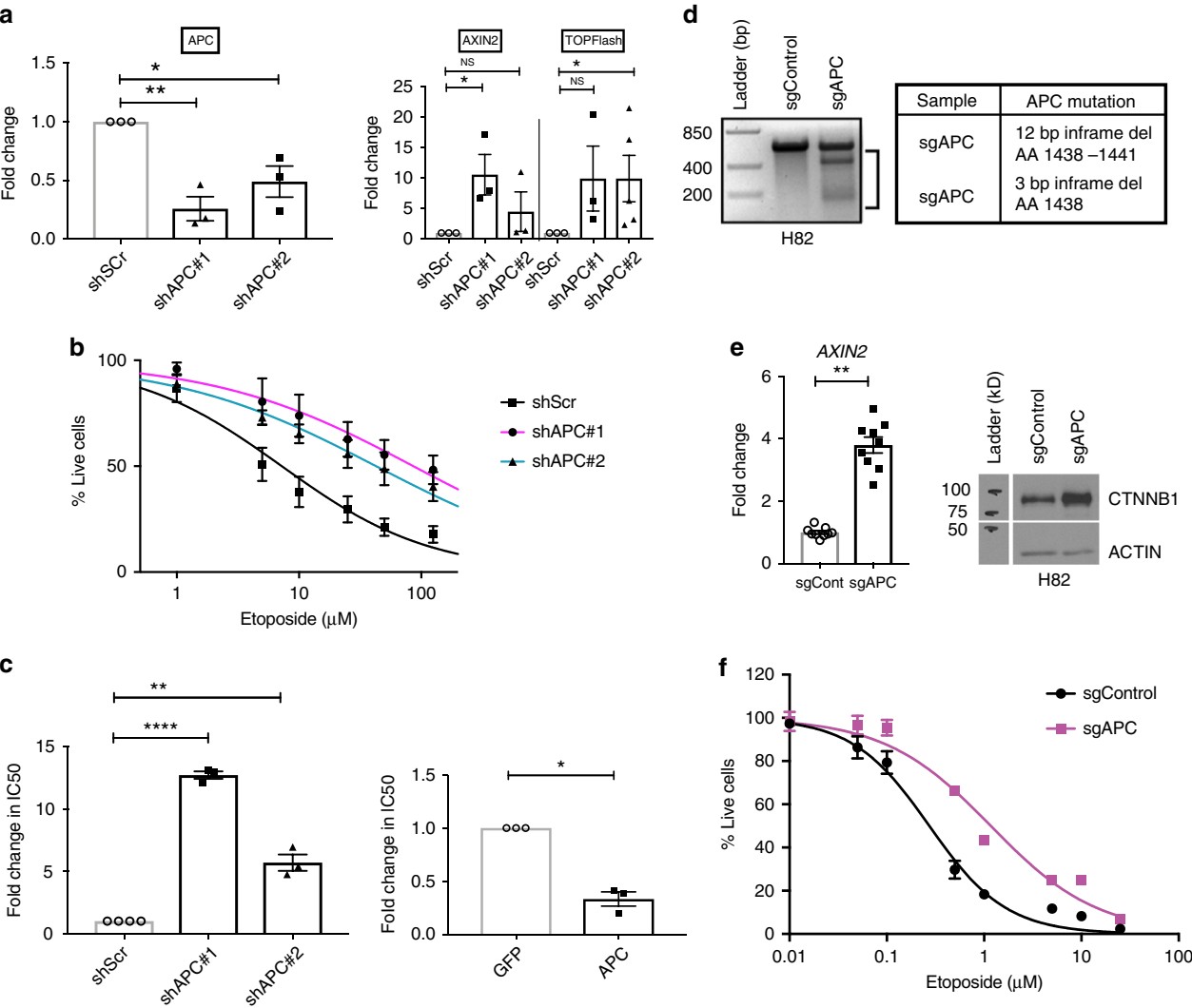

**Fig. 4** Loss of APC induces chemotherapy resistance in human SCLC cell lines. **a** Left: knockdown of *APC* in H1694 cells with two different shAPC constructs (shAPC#1 and shAPC#2); right: activation of WNT signaling as measured by *AXIN2* upregulation and TOPFlash reporter activity (signal fold changes) in these cells. Control cells expressed shRNA with scrambled target sequence (shScr). APC and AXIN2 mRNA levels were measured by quantitative PCR (qPCR). Fold change is reported with respect to control cells, and values were compared using unpaired *t* tests. Each experiment was performed in biological triplicate (TOPFlash assay results for shAPC#2 are shown from $n = 5$ experiments). **b** Percentage of H1694 cells surviving etoposide following 72-h treatment. **c** Left: fold change in etoposide IC50 following APC knockdown in H1694 cells compared to control cells, and right: fold change in etoposide IC50 in APC knockdown (shAPC#2) cells following overexpression of APC or GFP (control). IC50 values were compared using ratio-paired *t* tests. **d** Results from Surveyor assay demonstrating genomic alterations in *APC* (cleavage products indicated by black bar) following CRISPR–Cas9-guided deletion in H82 *sgAPC* cells and in-frame deletions in *APC* that were identified through targeted sequencing of the APC sgRNA site. **e** WNT activation in H82 *sgAPC* cells as measured by *AXIN2* mRNA levels by qPCR ($p < 0.01$, unpaired *t* test; $n = 3$ biological samples in technical triplicate) and CTNNB1 protein levels by immunoblot (representative of $n = 2$ independent experiments). Actin serves as loading control for CTNNB1. **f** Percentage of H82 cells surviving treatment with etoposide following CRISPR-guided deletion of APC (sgAPC) or control sequence (sgControl) ($n = 2$ experiments). Experiments were performed in biological triplicate, unless specified otherwise. IC50 inhibitory concentration 50, GFP green fluorescent protein, *$p < 0.05$, **$p < 0.01$, ****$p < 0.0001$, NS not significant. Bp base pair, Del deletion, AA amino acid, H1694 NCI-H1694, H82 NCI-H82. Error bars on box plots and dose-response curves are mean ± standard error

were reported for colorectal cancer by The Cancer Genome Atlas, where tumors harboring *APC* mutations were also found to harbor other WNT pathway alterations[38]. While the biological significance of cancer cells acquiring multiple alterations in the same pathway is unclear and may seem counterintuitive, it is possible that tumor cells select for these alterations to achieve an optimal level of WNT signaling[39]. On the other hand, these alterations could be confined to different subclones within the tumor and represent various mechanisms through which cancer

cells activate WNT signaling. We also observed LoH in negative regulators of canonical WNT signaling such as *APC* and *CSNK1A1* in nearly two-thirds of relapse SCLC samples. In conjunction with the high ssGSEA scores for WNT activation among relapse samples, this observation demonstrates that WNT signaling genes are likely to be altered through mechanisms other than non-synonymous mutations, which WES alone is unable to detect. Furthermore, components of the WNT signaling pathway are promiscuous in their interactions and demonstrate a great

degree of overlap with other pathways. Therefore, molecular studies of how alterations in WNT signaling affect the net activation of different pathways and a comprehensive analysis of the clonal architecture of tumors employing ultra-deep or single-cell sequencing techniques is likely to shed more light on the precise mechanisms of drug resistance in SCLC.

A recent study reported by Gardner et al. utilizing human SCLC patient-derived xenografts (PDXs) failed to observe mutational differences between treatment-naive and relapsed SCLCs[40]. EZH2-mediated epigenetic reprogramming that led to the silencing of *SLFN11* was identified as the mechanism driving induced chemotherapy resistance in this study—a finding which we were unable to reproduce, possibly due to a lack of paired primary-relapse gene expression data. Interestingly, another recent analysis by Drapkin et al., reported an association between increased expression of a *MYC* gene signature and chemotherapy resistance in SCLC PDXs derived from biopsy samples and circulating tumor cells, but failed to observe a correlation between *SLFN11* expression and chemotherapy sensitivity[41]. Heterogeneity in mechanisms that drive chemotherapy resistance in SCLC, sample characteristics, and differences in experimental design are likely factors contributing to the observed discrepancies between studies. For instance, since chemotherapy is known to affect immunogenic cell death in cancer cells and WNT signaling plays an important role in shaping tumor-host immune interactions, the utilization of immunodeficient mice for generating PDXs could have influenced the selection pressures acting on grafted tumors following exposure to chemotherapy in these studies[31,42].

To the best of our knowledge, this is the first study to perform comprehensive molecular characterization of paired human SCLC samples obtained at diagnosis and relapse. We acknowledge the limitations of our study, particularly with regard to small sample size, limited number of matched treatment-naive samples, and lack of RNA-sequencing data from treatment-naive samples. While these factors limit our ability to perform a number of additional informative analyses (e.g., tumor/relapse paired *CTNNB1* immunohistochemistry), it is extremely important to consider that patients with relapsed SCLC rarely undergo biopsies that yield enough tissue for high throughput sequencing. Our ability to demonstrate WNT pathway mutations and activation in relapse SCLC samples, and to validate their role in mediating chemotherapy resistance in vitro provide a rationale for studying WNT signaling inhibitors to overcome treatment resistance in SCLC.

## Methods

**Primary cohort sample acquisition and processing.** Tumor specimens for whole exome and transcriptome sequencing were obtained from patients diagnosed with SCLC who were treated at the Alvin J Siteman Cancer Center at Washington University School of Medicine. The Human Research Protection Office (HRPO) at the Washington University School of Medicine approved the study. All patients provided informed consent for participation and underwent a biopsy at the time of relapse and pre-treatment biopsy specimens were obtained for the twelve patients for whom a corresponding primary was available. Pathological diagnosis was confirmed by independent review by two pathologists (authors CJO and JHR; images available as Supplementary Data). Tumor specimens from multiple metastatic sites at relapse were available for two patients. For these patients, we also identified a single representative sample based upon observed VAFs for called mutations among all specimens from that patient. These representative samples were used in cohort-level analyses to maintain biological independence among tested samples in the cohort. Peripheral blood was used for germline analysis for all patients. Tissues were assigned a code and banked. Patient consenting, sample and data collection and storage were all conducted in compliance with institutional ethical regulations.

DNA and RNA extractions were performed using Qiagen kits. The DNeasy Blood and Tissue kit was used for DNA isolation from flash frozen (FF) materials. The QIAamp DNA FFPE Tissue or QIAamp DNA Mini kit was used for DNA isolation from FFPE materials. The RNeasy Mini kit was used for RNA isolation from FF materials.

Specimens were prepared in several batches (work orders) over the course of the study. Reference Supplementary Tables Sample Work Orders (Supplementary table 8) and Work Order Details (Supplementary table 9) for kits used for library construction and capture methods applied to each sample. Specimens were pooled together pre-capture, 400 ng/sample for a 1.6 μg input for capture. Each pool was sequenced over two lanes of an Illumina HiSeq sequencing instrument.

**Genome alignment and variant calling.** All genomic analyses were managed with the Genome Modeling System (GMS)[43]. Exome sequencing data were aligned to the human reference genome build GRCh37 using bwamem version 0.7.10 with default parameters[44]. Exome alignment models were processed by the GMS Somatic variation pipeline to identify single-nucleotide variants (SNVs), insertions and deletions (indels), and copy number alterations (CNAs)[43]. The union of snv/indel variant calls for each sample were manually reviewed using IGV[45] (Supplementary table 10). Somatic SNV detection was performed in this pipeline using a combination of SomaticSniper (version 1.0.2)[46], VarScan (version 2.3.6)[47], and Strelka (1.0.11)[48]. Somatic small indels were detected by a combination of GATK Somatic Indel Detector (version 5336)[49], Pindel (version 0.5)[50], VarScan, and Strelka. Annotation of variants was performed using the GMS transcript variant annotator against a GMS "annotation" build based on Ensembl v74_37 transcripts. SNVs and INDELs were further filtered by removing artifacts found in a panel of 905 normal exomes[51], removing sites that exceeded 0.1% frequency in the ExAC cohort release v0.2[52], and then using a bayesian classifier (https://github.com/genome/genome/blob/master/lib/perl/Genome/Model/Tools/Validation/IdentifyOutliers.pm) and retaining variants classified as somatic with a binomial log-likelihood of at least 10 (params: -llr-cutoff 10). Finally, these filtered mutations then underwent manual review with IGV.

**Tumor heterogeneity.** Variant allele frequencies were analyzed using SciClone 1.0.7 to visualize clonality[53]. Variants were visualized and compared with an *ad hoc* R script, available on the paper methods repository for this manuscript (https://github.com/ahwagner/sclc).

**Mutation signatures.** Mutation signatures were calculated using the deconstructSigs R package[10]. The signatures evaluated were the 30 COSMIC signatures for mutational processes in human cancers (http://cancer.sanger.ac.uk/cosmic/signatures). A difference of indel counts between limited and extensive stage samples (Supplementary table 1) was tested by unpaired *t*-test.

**Significantly mutated genes.** SMGs in relapsed samples were identified using MuSiC v0.4[13]. The region of interest file was created using regions annotated as CDS or RNA in Ensembl v74 with 2 bp flank added to contiguous exons in order to retain splice sites. Only relapse and normal samples were used for analysis. For individuals with multiple relapse samples, variants from all relapse samples for that individual were merged, deduplicated and used for analysis. A representative BAM file was also used for coverage calculations for these individuals. The "-merge-concurrent-muts" option was applied, allowing a single individual to only contribute, at most, a single mutation to the total mutation burden of any one gene. All other default parameters were used. The FDR cutoff was based on the convolution test (CT) method.

**Copy number alterations.** Segmentation files and marker files were generated for each sample using existing loss-of-heterozygosity data from genotyping arrays and the correct format was verified by comparison to the example data provided by the documentation. Using these files as input, GISTIC 2.0[16] was run using the GenePattern web module to call copy number alterations and evaluate cohort recurrence for significantly altered regions (https://genepattern.broadinstitute.org/gp). The supplied hg19 build of the reference genome was used. A germline CNA file was not provided. Default parameters were used for all sections under Additional Parameters and Advanced Parameters. Output file significant calls were verified by manual review using original BAM files in IGV.

**Loss of heterozygosity.** Heterozygous germline variant calls were selected from each patient sample where the normal VAF was within the range of 40–60. Absolute difference between the tumor VAF and 50 was calculated to produce a zygosity score. Segments were generated using the DNAcopy[54] circular binary segmentation algorithm, and average zygosity score of constituent variants attributed to the segments. Biomart[55] was used to obtain genomic coordinates for each gene and overlap with the segment coordinates, to generate a zygosity score for each gene in each sample. A zygosity score above 30 (representing an average tumor VAF < 20 or >80) was used to call a gene as loss-of-heterozygosity (LoH) (Supplementary table 11). In cases where there was insufficient data to make a segment, the zygosity score was omitted, and no LoH call was made.

**Gene sets.** Cancer relevant genes were collected from Supplementary table 2 of Lawrence et al., and filtered for a pan-cancer *q* value < = 0.1, as discussed in the main text of that paper (list of genes available in Supplementary table 6). A list of genes participating in canonical WNT signaling was curated from Reactome

(Supplementary table 7) after analyzing WNT gene lists R-HSA195721 (Signaling by WNT) and R-HSA-3858494 (β-catenin independent WNT signaling).

**Transcriptome alignment and expression estimation**. Fasta files were trimmed using Flexbar[56] with settings: -adapter-min-overlap 7 -min-read-length 20 to improve read alignments. Reads were aligned by hisat2.0.5[57] using Ensembl version 74 transcripts. Reads were sorted and indexed by SAMtools 1.3.1[58] and then used for expression estimation by Stringtie 1.3.3[59].

**Expression batch correction and sample clustering**. Stringtie expression estimates were loaded by Ballgown[60], expression filtered, log transformed, normalized by ComBat[61], and clustered with pheatmap (https://github.com/raivokolde/pheatmap) in an ad hoc script available on the methods repository for this manuscript (https://github.com/ahwagner/sclc).

**ssGSEA**. Expression estimates were combined with human primary tumor expression data from George et al.[3]. Data were then manually transformed into a GCT file:
(http://software.broadinstitute.org/cancer/software/genepattern/file-formats-guide#_Creating_Input_Files_Tab) and run through ssGSEA v9.0.6 in GenePattern (https://genepattern.broadinstitute.org). Relevant gene sets were selected from MutSigDB (http://software.broadinstitute.org/gsea/msigdb/genesets.jsp) and downloaded as GMT files. The input and result files (Supplementary table 12) of this analysis are available on the methods repository for this manuscript (https://github.com/ahwagner/sclc).
ssGSEA was validated by calculating ssGSEA enrichment scores using an ASCL1 expression signature (as defined by Borromeo et al.)[5], and exploring the correlation between ASCL1 enrichment and tumor subtype (ASCL1-high [A], NEUROD1-high [N], and Dual negative [D]). Subtype information for George et al. treatment-naive samples was obtained from the publication by Mollaoglu et al.[6] Finally, independently categorized samples were grouped together by subtype for this analysis, to test the ability of ssGSEA to detect true biological differences independent of batch effect. As expected, ASCL1 high tumors from both studies were most enriched for the ASCL1 signature, while dual negatives were the least enriched (one-way ANOVA analysis; $p < 0.0001$). Similarly, analyses by Mollaoglu et al. and Polley et al. demonstrated high MYC activity in ASCL1-low SCLC samples and cell lines[6,62]. Using the ssGSEA method, we calculated enrichment scores for MYC activation (using the MYC activation signature defined by Bild et al.) and ASCL1 driven gene expression (defined by Borromeo et al.) and examined their correlation[5,62]. Once again, independently subtyped SCLC samples were grouped together for this analysis. As expected, MYC activation was inversely correlated with high ASCL1 enrichment scores (linear regression; $p = 0.003$).

**Primers**

| Primer | Sequence |
| --- | --- |
| H82, DMS53, and H1048 cell line experiments | |
| sgControl Fwd | CACCGGCGAGGTATTCGGCTCCGCG |
| sgControl Rev | AAACCGCGGAGCCGAATACCTCGCC |
| Human sgAPC Fwd | CACCGGTTTGAGCTGTTTGAGGAGG |
| Human sgAPC Rev | AAACCCTCCTCAAACAGCTCAAACC |
| Actin Fwd | TATTGGCAACGAGCGGTTCC |
| Actin Rev | GGCATAGAGGTCTTTACGGATGTC |
| AXIN2 Fwd | GCCAAGTGTCTCTACCTCATT |
| AXIN2 Rev | TTTCCAGCCTCGAGATCA |
| APC Fwd | CAGATTCTGCTAATACCCTGCAA |
| APC Rev | CCATCTGGAGTACTTTCCGTG |
| H1694 cell line experiments | |
| AXIN2 Fwd-2 | CGGACAGCAGTGTAGATGGA |
| AXIN2 Rev-2 | CTTCACACTGCGATGCATTT |

**H82, H1048, and DMS53 cell culture and viral preparation**. NCI-H1048 and DMS53 cells (kindly provided by David MacPherson) were maintained in RPMI. H82 cells were obtained from ATCC and maintained in RPMI. All medias contained 10% FBS, 1% Pen/Strep and 1% L-glutamine. Cell lines were validated by STR profiling at the University of Utah DNA Sequencing Core Facility using the GenePrint 24 kit from Promega (Madison, WI, USA) in June and December 2017. Chemotherapy resistant clones of these cell lines were derived from parental lines through repeated in vitro cisplatin and etoposide treatment followed by serial passaging. Targeted Sanger sequencing of *APC* was performed on these lines.
For generating viral supernatants, 293T cells were transfected using Mirus LT-1 transfection reagent with gene plasmids as indicated along with Gag/pol and Env plasmids. Supernatants were harvested after 48–72 h and cell lines were infected

twice with viral supernatant at 1:1 (media/supernatant) with 1000× polybrene (8 µg/ml). Cells were then selected for 2–3 days in puromycin.
Validated human APC sgRNA[63] and sgControl primers were subcloned into Lenti-CRISPR-V2[64] (Addgene plasmid #52961). H82 cells were infected with Lenti-V2-sgAPC or sgControl lentivirus and stable polyclonal pools were selected with puromycin.

**H82, H1048, and DMS53 quantitative real-time PCR**. For gene expression analysis by real-time PCR, RNA was isolated by TRIzol from Invitrogen (Carlsbad, CA, USA) or RNeasy kit from Qiagen (Hilden, Germany) and 1 µg of total RNA was converted to cDNA using iScript cDNA synthesis kit from Bio-Rad (Hercules, CA, USA). Quantitative real-time PCR was performed using gene-specific primers and Sybr Green Supermix (Bio-Rad) in a 20 µl reaction in triplicate on a Bio-Rad CFX96 Real-Time PCR machine. Analysis was performed using Bio-Rad CFX Manager software and expression values were based on 10-fold serial dilutions of standards and normalized to Actin levels. Actin Fwd and Rev, and AXIN2 Fwd and Rev primers were used for real-time PCR.

**H82, H1048, and DMS53 cell viability assays**. Cells were seeded in triplicate (5000/well) in opaque 96-well plates. The next day, cells were treated with increasing doses of cisplatin (0–100 µM), or etoposide (0–100 µM) obtained from Sigma (St. Louis, MO, USA). After 48–96 h of treatment, cell viability was measured using CellTiter Glo from Promega (Madison, WI, USA) on a luminometer. Normalized, transformed dose response curves were generated and analyzed using GraphPad Prism (GraphPad Software Inc.; LaJolla, CA, USA) to determine IC50.

**H82, H1048, and DMS53 immunoblotting**. Total protein lysates or nuclear and cytoplasmic fractions were separated via SDS-PAGE and transferred to a PVDF membrane. Membranes were blocked for 1 h, followed by overnight incubation with primary antibodies from Cell Signaling Technologies (Danvers, MA, USA): B-Catenin (9587S, Carboxy terminal antigen) or PARP (46D11) at 1:1000 dilution, B-Tubulin (Developmental Studies Hybridoma Bank, 1:200) or Beta-Actin (#A2066, 1:10,000) from Sigma-Aldrich (St. Louis, MO, USA) at 4 ℃. Membranes were washed at room temperature in TBS-T. Mouse or rabbit HRP-conjugated secondary antibodies from Jackson ImmunoResearch (West Grove, PA, USA) at 1:10,000 dilution were incubated for 1 h at room temperature. For detection, membranes were exposed to WesternBright HRP Quantum substrate from Advansta (Menlo Park, CA, USA) and detected on Hyblot CL film from Denville Scientific Inc. (Holliston, MA, USA). Uncropped blots are available in Supplementary Information (Supplementary fig. 6).

**H82 surveyor assay and sequencing**. Genomic DNA was isolated from snap-frozen cell pellets with the DNeasy kit from Qiagen. Human APC PCR products were amplified with APC Fwd and APC Rev primers. sgRNA-induced alterations were detected with the IDT Surveyor Mutation Detection Kit (Integrated DNA Technologies, Coralville, IA, USA; #706030) following the manufacturer's specifications. Amplified APC PCR products were subcloned into TOPO vectors from Life Technologies (Carlsbad, CA, USA) and individual colonies were sequenced to identify APC alterations.

**H1048 and DMS53 RNA sequencing**. RNA was extracted from cells using RNeasy Kit from Qiagen. The RNA-seq libraries were generated using Illumina TruSeq Stranded mRNA sample preparation kit with oligo dT selection. Sequencing libraries (25 pM) were chemically denatured and applied to an Illumina HiSeq v4 paired end flow cell using an Illumina cBot. Hybridized molecules were clonally amplified and annealed to sequencing primers with reagents from an Illumina HiSeq PE Cluster Kit v4-cBot. Following transfer of the flowcell to an Illumina HiSeq instrument at the University of Utah High-Throughput Genomics Core, a 125 cycle paired-end sequence run was performed using HiSeq SBS Kit v4 sequencing reagents with six samples multiplexed in each Illumina lane. The average number of reads per sample ranged from 38 to 58 million reads.

**H1048 and DMS53 gene expression analyses**. Gene expression was determined using the RSEM utility "rsem-calculate-expression" using the stranded and paired-end options. Differential expression was determined using EBSeq (v1.4.0) using "MedianNorm" function to calculate size factors and setting "maxround" to 10 (ref. [55]). For hierarchical clustering based on sample distances, counts were loaded into the edgeR (v3.8.5) DGEList object. Genes with counts-per-million (cpm) greater than 2 in at least two samples were kept. Counts were normalized using the edgeR calcNormFactors method with default settings. A matrix of log2 cpm values was generated using the edgeR function predFC setting prior.count to 2. The Euclidean distances calculated using the transposed log2cpm matrix were run through heatmap.2 using default settings. Differentially expressed genes (FC > 1.5 or 2, FDR < 0.05) were analyzed for enrichment of pathways, tissue expression and biological processes using QIAGEN's Ingenuity Pathway Analysis (IPA, QIAGEN).

**H1694 cell culture and virus production**. Small cell lung cancer cell line NCI-H1694 was a gift from the Oliver lab (author TO). STR testing was performed in

the Oliver lab to verify cell line identity. Cell lines were maintained in RPMI 1640 media supplemented with 10% FBS. Cells were maintained at 37 °C in 5% $CO_2$. For lentiviral production, $5 \times 10^6$ 293T cells were co-transfected with pCMV-VSV-G, pCMVΔR8.2, and pLKO.1-puro constructs using Lipofectamine 2000 from ThermoFisher (Waltham, MA, USA). Forty-eight hours post transfection, viral supernatants were collected and pooled. NCI-H1694 cells were incubated with viral supernatant for 24 h, in the presence of polybrene (10≥μg/mL). Cells were then selected with puromycin for 48 h.

**APC knockdown and overexpression in H1694 cells.** APC (NM_000038.2) knockdown was performed using pLKO.1-shRNA constructs obtained from the McDonnell Genome Institute at Washington University. At least two shRNAs targeting APC at different locations were utilized in each knockdown experiment. shRNA construct #1 (seed sequence: CCCAGTTTGTTTCTCAAGAAA) targeted positions 6136–6156 in exon 16 (NM_000038.5), while #2 (seed sequence: TAATGAACACTACAGATAGAA) targeted nucleotides 8921–8941 in the 3′ UTR (mRNA transcript number NM_000038.5 in both instances). Stable knockdown of APC in H1694 cells was achieved through lentiviral transduction of shRNA expressing virus, followed by puromycin selection. The pLKO.1-scramble shRNA (Addgene, plasmid #1864) was used as a knockdown control for these experiments.

To study the effects of restored APC expression on cell viability in the presence of cisplatin or etoposide, the pcDNA3.1-APC construct (lacking UTRs) was transfected into H1694 cells stably expressing shAPC#2 (sequence targets region 8921–8941, which lies in 3′ UTR). Plasmid with APC open reading frame (NM_001127510.2 ORF) in a pcDNA3.1 backbone was obtained from Genscript (Piscataway, NJ, USA; catalog number OHu16786) for APC overexpression experiments. GFP (eGFP-C1 vector, GenBank accession# U55763) was overexpressed in these cells to serve as a control for APC overexpression in these experiments. shAPC#2 cells were plated with chemotherapy for viability assessment 48 h after transduction with Lipofectamine 2000 from ThermoFisher (Waltham, MA, USA).

**Quantitative real-time PCR assays in H1694 cells.** Total RNA was isolated from transduced cells using the NucleoSpin RNA kit from Machery-Nagel (Bethlehem, PA, USA) according to the manufacturer's instructions. Three micrograms of total RNA was used to make cDNA using the SuperScript II 1st Strand Synthesis Kit from Life Technologies (Carlsbad, CA, USA) with oligo dT primer according to the manufacturer's instructions. Real-time PCR was performed on a CFX96 Real-Time PCR machine manufactured by Bio-Rad (Hercules, CA, USA) using iTaq Universal SYBR Mix. Predesigned qPCR assays for APC (Hs.PT.56a.3539689) and β-Tubulin (TUBB)(Hs.PT.58.15482507) were purchased from Integrated DNA Technologies (Coralville, IA, USA). Custom primers for AXIN2 (AXIN2 Fwd-2 and Rev-2 primers) were used for measuring WNT activation and were also purchased from IDT[65]. TUBB was used as a normalization reference gene for all assays.

**H1694 cell viability assays.** NCI-H1694 with APC knockdown (shAPC#1, shAPC#2) and shAPC#2 cells overexpressing APC, and their appropriate controls (shScr for knockdown and GFP for overexpression experiments) were plated in 96 well plates in RPMI medium containing 10% FBS, with cisplatin obtained from Selleck chemicals (Houston, TX, USA), diluted in DMF, for 48 h and etoposide, diluted in DMSO (Selleck chemicals), for 72 h. Cell viability at the end of each incubation period was measured using alamarBlue (catalog number DAL1025) obtained from ThermoFisher (Waltham, MA, USA) according to manufacturer's instructions. Briefly, cells were incubated with alamarBlue for 24 h, after which alamarBlue reduction was measured by fluorescence on a Spectramax M5 plate reader manufactured by Molecular Devices (Sunnyvale, CA, USA).

**H1694 TOPFlash assay.** WNT activation was assessed by TOPFlash assay[66], after optimizing the experimental conditions for NCI-H1694 cells. WNT-responsive firefly luciferase reporter plasmid with 7 TCF/LEF binding sites (M50 Topflash; plasmid #12456), renilla luciferase (pMULE-Renilla-CMV; plasmid #62186), and mutant TOP-Flash (M51-FOPFlash; Plasmid #12457) plasmids were obtained from Addgene (Cambridge, MA, USA). Briefly, $1 \times 10^6$ cells lentivirally transduced with shAPC and shScr were plated in six-well plates with media containing 10% FBS, following puromycin selection. Cells were then co-transduced with TOPFlash DNA and Renilla luciferase (10:1 ratio) using Lipofectamine 2000 from ThermoFisher (Waltham, MA, USA). Co-transduction of FOPFlash (which contains mutant TCF binding sites) and Renilla luciferase in a similar 10:1 ratio was performed as a control for this experiment. Media was exchanged after 24 h. Luciferase assay readings were obtained using the Dual Glo Luciferase assay kit from Promega on a TD-20/20 Luminometer manufactured by Turner Designs (Sunnyvale, CA, USA), according to manufacturer's instructions at 48 h post transfection. Ratio of FOP-Flash/Renilla luciferase values was subtracted from the ratio of TOPFlash/Renilla luciferase values.

**H1694 statistics.** All experiments were performed at least in triplicate and GraphPad Prism 7 (GraphPad Software Inc., LaJolla, CA, USA) was used for performing statistical analyses. IC50 values for chemotherapy agents was obtained from cell survival assays using the variable slope model with least squares fit. qPCR expression fold changes for *APC*, *AXIN2*, and TOPFlash WNT reporter values (Fig. 4a) were compared using Student's *t* test. IC50 fold change values in different experiments were compared using ratio-paired *t* tests (Fig. 4c, Supplementary figs. 4b, c).

**ASCL1 immunohistochemistry in human SCLC validation cohort.** We obtained an independent set of 12 pre-treatment and 10 post-treatment human SCLC samples from patients who received standard platinum chemotherapy. Excess deidentified formalin-fixed paraffin embedded tissue was obtained from the Intermountain BioRepository and Pathology Departments of Intermountain Healthcare under protocol approved by the institutional review board (#1040177). Longitudinal clinical patient data were extracted from the Intermountain electronic data warehouse. Samples and data were de-identified by the Intermountain BioRepository data manager. Institutional guidelines regarding specimen and data use were followed.

Paraffin-embedded lung lobes were sectioned at 4 μm and stained with H&E for tumor pathology. Sections were dewaxed, rehydrated and subjected to high temperature antigen retrieval, 20 min boiling in a pressure cooker in 0.01 M citrate buffer, pH 6.0. Slides were blocked in 3% $H_2O_2$ for 15 min, in 5% goat serum in PBS/0.1% Tween-20 for 1 h, and stained overnight in 5% goat serum in PBS/0.1% Tween-20 with primary antibodies. HRP-conjugated secondary antibody was used at 1:200 dilution in 5% goat serum in PBS/0.1% Tween-20, incubated for 45 min at RT, followed by treating tissues with ABC reagent for 30 min and DAB staining (Vector Laboratories). All staining was performed with Sequenza coverplate technology. ASCL1 antibody (clone 24B72D11, 1:200 dilution) was obtained from Fisher Scientific (Waltham, MA, USA).

**Code availability.** Custom code and intermediate results for these analyses are available at GitHub (https://github.com/ahwagner/sclc) under the MIT license (no restrictions on reuse).

## Data availability

Data has been deposited in dbGAP with accession #PHS001049 (https://www.ncbi.nlm. nih.gov/gap/?term=PHS001049), and SRA with project accession PRJNA306801 (https:// www.ncbi.nlm.nih.gov/bioproject/?term=PRJNA306801).

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

## Acknowledgements

The authors would like to thank Erica Barnell, Irena Lanc, and Lynzey Kujan for their assistance with manual review of variants. A.H.W. was supported by a fellowship (F32CA206247) and traineeship (T32CA113275) from the NCI. S.D. was supported by a fellowship award from the International Association for Study of Lung Cancer (IASLC). T.G.O. was supported by the Huntsman Cancer Foundation (P30CA042014) and in part by the NIH (R01CA187487, R21CA216504) and the American Lung Association (LCD-506758). M.G. was supported by the NHGRI under award R00HG007940. O.L.G. was supported by the NCI under award numbers K22CA188163 and U01CA209936. R.G. was supported by a grant from the Lung Cancer Connection foundation.

## Author contributions

A.H.W., Z.L.S. and A.R. performed GMS analyses. D.M., S.N.W. and R.G. clinically evaluated patients, contributed clinical data, and tumor samples. R.S.F. was responsible for sequencing strategy design. J.W. and J.P. performed library prep and sequencing. SMG analysis was performed by K.K. Z.L.S. performed mutation signature analysis. S.D., L.B.M., J.D.W. and T.G.O. designed lab experiments. S.D., R.R.O., H.C., A.M., M.H.C. and I.C. performed lab experiments. L.B.M. and T.G.O. supervised lab experiments. A.H. W. S.D., R.R.O. and Z.L.S. generated figures for manuscript. Manual review of variants was done by AHW, A.M., J.K., L.T. Copy number analysis was performed by J.K. and N.C.S. A.H.W. and S.D. performed RNAseq analyses and interpretation. Loss of heterozygosity analysis was performed by J.K., A.H.W., C.J.O. and J.H.R. reviewed histopathology images for cohort. E.R.M., R.K.W., M.A.W., M.G., O.L.G., T.G.O. and R.G. designed the project. O.L.G., T.G.O. and R.G. supervised the project. S.D., A.H.W., K.K., D.M., S.N.W., L.B.M., R.R.O., E.R.M., T.G.O., O.L.G. and R.G. edited the manuscript. All authors approved the manuscript.

## Additional information

**Competing interests:** The authors declare no competing interests.

