## [Peer Review File · Nature Communications]

Reviewers' comments:

Reviewer #1 (Remarks to the Author):

NCOMMS-18-03586-T

Wagner et al. "Recurrent WNT Pathway Alterations are Frequent in Relapsed Small Cell Lung Cancer "

The authors perform whole exome sequencing on small cell lung cancer (SCLC) tumor samples acquired either as 12 pairs (before and after chemotherapy/radiation therapy) or in 18 "relapsed" tumor samples. They also performed RNASeq on 18 relapsed tumor samples. Overall, the tumor mutation burden (TMB), mutational carcinogen signatures, genes found mutated, and RNA expression profiles corresponded to prior reports. The main findings relating to changes in relapsed patients were the enrichment for lineage oncogene ASCL1 negative expression patterns (change from 6/12 ASCL1+ pretreatment to 0/10 ASCL1+ post treatment), the discovery of some patterns of mutated genes not previously associated with SCLC such as COL11A1, deletions of MYCL, and mutations in WNT pathway genes. Overall they found 603 genes to have mutations in relapsed samples not previously found in untreated samples and through a series of computational analyses focused on several genes in the WNT pathway that were altered in some 80% of the relapsed samples and 50% of the paired samples at relapse. They performed studies of a 2-3 SCLC human line preclinical models to show that knockdown of a WNT pathway gene APC led to etoposide resistance and SCLC lines selected to be resistant exhibited increased WNT pathway activity. They have a brief discussion relating their findings to a recently published finding relating EZH2 driven suppression of SLFN11 expression as a key mechanism of resistance in SCLC. They conclude: "Our results support a role for WNT signaling activation as a mechanism of chemotherapy resistance in relapsed SCLC. "

Comments to the Authors:

The manuscript is reviewed in the context of the urgent need to obtain information about the mechanisms of resistance to standard chemotherapy in SCLC that would allow the development of new ways to prevent or overcome this resistance to provide new major steps forward in therapeutic responses. The most important part of this study is the acquisition of the patient tumor specimens that allowed this study to occur. The molecular analyses (exome seq and RNA-seq) are technically well done and the computational biology also competently done by one of the major genomics centers. However, I am sure the authors agree these laboratory analyses could be performed by just about any center with experience in this area. I believe the findings presented. However, there are some major problems with the presentation of the data the authors need to correct.

1. The key patient data in Table S1 shows two things. The first is that there are no data given on the standard clinical patient responses to initial therapy and the duration of this response. As part of this the therapy is listed as a compendium of therapies for nearly all patients and thus it must be the case these were a sequence of different therapies given to the patients. Of course, this means we need to know when in this series of therapies the relapsed biopsies were obtained. The paper as presented suggests that there were biopsies before treatment (true) and then after treatment at relapses (much more complicated). They need to provide this the timing of the biopsies after the various therapies. We could assume the relapsed biopsies came after all of the treatment was given. In this case that means 21/30 patients had much more complex treatments than the combination of platin + etoposide with or without irinotecan. Of course this means that any mutational pattern could be quite different than that for developing resistance to platin-etoposide – standard front line therapy. In any event, in the current version of the paper this aspect is to say the least not presented clearly and needs to be dealt with.

2. The authors need to go over all of their figure legends and figures. I found multiple examples where things were not presented as listed or not presented clearly. Some examples are in Figure 3

there is no MYCL staining shown although this is stated in the legend; in Figure 4 panel A not clear what is shown since legend states over expression of APC and not at all clear where that is, and in addition, what the Y axis shows is not clear.

3. The claim of preclinical model resistance after altering APC is not well presented. What is needed is a simple etoposide dose response curve with and without shRNA knockdown. In addition, the amount of resistance after CRISPR knockout appears to be quantitatively very small and the actual fold changes (which appear, as 2-4 fold at most needs to be stated.

4. Finally, coming back to point 1 above, how do we know all of the patients indeed have a diagnosis of SCLC? While the dual TP53, and RB1 mutations and ASCL1 or NEUROD1 expression would point to that in many cases, they actually found a lot of their tumors were wild type for TP53 and RB1 and of course they have the many that were ASCL1 and neuroendocrine negative. This publication comes from a major Center so these probably are SCLC cases – but they need to present straightforward data validating that point for each patient. One simple way would be to have the H&E stained image for each patient available as supplemental data and then some summary columns that list key features validating that the tumors are SCLC such as TP53 status, RB1 Status, ASCL1 status, NEUROD1 status, other neuroendocrine markers, and clinical presentation (stating either “typical” or “atypical” for SCLC).

5. The relapsed samples came from several different metastatic sites (I assume that Suppl Table 1, column L that is titled “Tissue Description”) and companion Column I is where the samples came from. Was there any difference in the mutational pattern dependent on where the relapsed biopsy came from?

6. From the end of the paper I was not clear whether they felt their findings agreed or disagreed with the EZH2/SLFN11 findings. Whatever the result is (that some do and others don’t or something else) this needs to be clearly stated.

7. I assume all of the genomics data will be deposited as part of this publication and that, of course is an iron-clad requirement.

Reviewer #2 (Remarks to the Author):

The authors carried out whole-exome sequencing analyses of relapsed small cell lung cancer (SCLC) to elucidate its mechanisms. They found loss-of-function alterations in the APC gene in relapsed SCLC cases, as well as chemotherapy resistant SCLC cell lines, and underlined involvement of the canonical WNT/ β -catenin signaling activation in relapsed or chemotherapy-resistant SCLCs.

The following points were noticed as issues to be further addressed:

(1) Flow of manuscript. The authors showed whole spectrum of genetic alterations in relapsed SCLCs in Figure 1, whereas in the following parts they focused on relatively rare events related to genetic alterations in canonical WNT signaling components and mechanistic analyses to explain how the canonical WNT signaling activation leads to relapse or chemotherapy resistance. I am afraid that the flow of this manuscript was hindered between Figure 1 and the following parts.

(2) Precise contribution rate of canonical WNT/ β -catenin signaling in SCLC relapse. The authors detected alterations of the APC gene (N507I, Q1204*, H1232Y and E1464fs) in 4 of 30 cases of relapsed SCLCs. Q1204* and E1464fs are loss-of-function APC alterations, whereas functional consequences of N507I and H1232Y APC variations remain unclear. These facts indicate that loss-of-function APC alterations occur in 6.7 % (2 / 30) of relapsed SCLC cases. By contrast, the canonical WNT/ β -catenin signaling cascade is aberrantly activated in tumor cells owing to coding and non-coding alterations in the APC, AXIN1, AXIN2, CTNNB1, RNF43, RSPO2, RSPO3 and ZNRF3 genes as well as canonical WNT ligand elevation in the tumor microenvironment. The precise contribution rate of canonical WNT/ β -catenin signaling activation in SCLC relapse is

underestimated in this exome sequencing-based study. The authors are advised to address this issue to improve the value of this report for diverse groups of clinical oncologists.

(3) Lack of omics data other than genomics and transcriptomics data. This report mainly depends on genomics and transcriptomics data. Related to (2), the authors need to carry out immunohistochemical analysis of β -catenin to strengthen their claim on the contribution of the canonical WNT/ β -catenin signaling in relapse or chemotherapy-resistance of SCLC.

Reviewers' comments:

Reviewer #1 (Remarks to the Author):

The authors perform whole exome sequencing on small cell lung cancer (SCLC) tumor samples acquired either as 12 pairs (before and after chemotherapy/radiation therapy) or in 18 “relapsed” tumor samples. They also performed RNASeq on 18 relapsed tumor samples. Overall, the tumor mutation burden (TMB), mutational carcinogen signatures, genes found mutated, and RNA expression profiles corresponded to prior reports. The main findings relating to changes in relapsed patients were the enrichment for lineage oncogene ASCL1 negative expression patterns (change from 6/12 ASCL1+ pretreatment to 0/10 ASCL1+ post treatment), the discovery of some patterns of mutated genes not previously associated with SCLC such as COL11A1, deletions of MYCL, and mutations in WNT pathway genes. Overall they found 603 genes to have mutations in relapsed samples not previously found in untreated samples and through a series of computational analyses focused on several genes in the WNT pathway that were altered in some 80% of the relapsed samples and 50% of the paired samples at relapse. They performed studies of a 2-3 SCLC human line preclinical models to show that knockdown of a WNT pathway gene APC led to etoposide resistance and SCLC lines selected to be resistant exhibited increased WNT pathway activity. They have a brief discussion relating their findings to a recently published finding relating EZH2 driven suppression of SLFN11 expression as a key mechanism of resistance in SCLC. They conclude: “Our results support a role for WNT signaling activation as a mechanism of chemotherapy resistance in relapsed SCLC. “

Comments to the Authors:

The manuscript is reviewed in the context of the urgent need to obtain information about the mechanisms of resistance to standard chemotherapy in SCLC that would allow the development of new ways to prevent or overcome this resistance to provide new major steps forward in therapeutic responses. The most important part of this study is the acquisition of the patient tumor specimens that allowed this study to occur. The molecular analyses (exome seq and RNA-seq) are technically well done and the computational biology also competently done by one of the major genomics centers. However, I am sure the authors agree these laboratory analyses could be performed by just about any center with experience in this area. I believe the findings presented. However, there are some major problems with the presentation of the data the authors need to correct.

1. The key patient data in Table S1 shows two things. The first is that there are no data given on the standard clinical patient responses to initial therapy and the duration of this response. As part of this the therapy is listed as a compendium of therapies for nearly all patients and thus it must be the case these were a sequence of different therapies given to the patients. Of course, this means we need to know when in this series of therapies the relapsed biopsies were obtained. The paper as presented suggests that there were biopsies before treatment (true) and then after treatment at relapses (much more complicated). They need to provide this the timing of the biopsies after the various therapies. We could assume the relapsed biopsies came after all of the treatment was given. In this case that means 21/30 patients had much more complex treatments than the combination of platin + etoposide with or without irinotecan. Of course this means that any mutational pattern could be quite different than that for developing resistance to platin-etoposide – standard front line therapy. In any event, in the current version of the paper this aspect is to say the least not presented clearly and needs to be dealt with.

We agree with this comment from the reviewer and have updated supplementary table 1 to include information on the time points at which relapsed samples were obtained in relation to the patient's treatment history. To summarize the findings of that table, all patients underwent standard front-line therapy with cisplatin/carboplatin and etoposide (c & e treatment) and often with concurrent or subsequent radiation therapy. These cases were all followed by one or more additional treatments after relapse. The majority of the cohort had the biopsy taken at first relapse after standard front-line therapy (18/30). Specifically, the biopsy was taken after only c & e treatment in 7 / 30 patients, with an additional 11 having also received radiation therapy. Biopsies from 12 patients were taken after treatment with one or more additional drugs beyond c & e (such as irinotecan or topotecan). Time to relapse from the date of initial diagnosis is listed in Supplementary Table S1, column E.

2. The authors need to go over all of their figure legends and figures. I found multiple examples where things were not presented as listed or not presented clearly. Some examples are in Figure 3 there is no MYCL staining shown although this is stated in the legend; in Figure 4 panel A not clear what is shown since legend states over expression of APC and not at all clear where that is, and in addition, what the Y axis shows is not clear.

We appreciate the referee for their attention to detail and apologize for the lack of clear descriptive captions in our initial submission. We have made extensive modification to the figure layouts and caption text to address these concerns. Broadly, we have standardized our nomenclature and styles across figures where applicable, including coloring, indication of significance, reported units (e.g. fold change) and representation of variance. Subfigures have been split into additional panels in Figures 2, 3, and 4 for clarity of discussion (highlighted in legends, lines 264-274, 277-283, 285-302). The specific issues highlighted by the referee have also been carefully reviewed and addressed. The results text discussing MYCL expression now directly references Figure 3c (lines 161-162), and the figure legend now correctly indicates that MYCL expression (not staining) is indicated in panel c (line 281). Additional clarifying text indicating that representative staining in panel b is for ASCL1 has also been added (line 279). Figure 4 panel A has been revised to describe knockdown of APC expression as measured by qPCR (lines 289-290) and overexpression of WNT signaling by *AXIN2* upregulation and TOPFlash reporter activity (lines 287-288). The Y axis in Figure 4a still indicates fold change, as in our initial submission, but we have added additional clarifying detail that the assay (qPCR / TOPFlash, respectively) fold change is with respect to the control cells (line 290).

3. The claim of preclinical model resistance after altering APC is not well presented. What is needed is a simple etoposide dose response curve with and without shRNA knockdown. In addition, the amount of resistance after CRISPR knockout appears to be quantitatively very small and the actual fold changes (which appear, as 2-4 fold at most needs to be stated.

We have modified Figure 4 to include an etoposide dose response curve (Figure 4b) with and without shRNA knockdown, to supplement the existing sgAPC dose-response curve in Figure 4f. These curves demonstrate a significant increase in etoposide resistance with the knockdown of APC. The IC50 fold change for the CRISPR knockout is 4.341x (0.270 vs 1.172 μ M). We explicitly reference the CRISPR knockout in the manuscript (lines 197-201), and there have added the following text in response to the referee suggestion:

The extent to which these cells became etoposide resistant was modest in comparison to H1694 cells with APC knockdown, with a 4.3-fold increase in IC50 (**Figure 4f**).

4. Finally, coming back to point 1 above, how do we know all of the patients indeed have a diagnosis of SCLC? While the dual TP53, and RB1 mutations and ASCL1 or NEUROD1 expression would point to that in many cases, they actually found a lot of their tumors were wild type for TP53 and RB1 and of course they have the many that were ASCL1 and neuroendocrine negative. This publication comes from a major Center so these probably are SCLC cases – but they need to present straightforward data validating that point for each patient. One simple way would be to have the H&E stained image for each patient available as supplemental data and then some summary columns that list key features validating that the tumors are SCLC such as TP53 status, RB1 Status, ASCL1 status, NEUROD1 status, other neuroendocrine markers, and clinical presentation (stating either “typical” or “atypical” for SCLC).

We thank the referee for their diligence in considering the diagnosis of SCLC for the patients described in this manuscript. All patients were diagnosed with SCLC at the Washington University in St. Louis by board certified pathologists. Additionally, pathology was independently reviewed from H&E images by author J.R. All images have been added to the supplement as “Supplementary Data 1.zip”. We have also added an immunohistochemical description of samples where available as Supplementary Table 2. IHC for ASCL1 and NEUROD1 was not performed and samples were not classified as typical or atypical since this is not standard clinical practice. While Figure 3 of our manuscript describes ASCL1 staining, this was in independent validation samples that are distinct from the 30 SCLC patients described earlier in the manuscript.

With regard to canonical genomic alterations in this cohort, an independent study and review of the genomic landscape of primary SCLCs reported that point mutations in TP53 and RB1 have been observed in 90-95% and 65%-80% of SCLCs, respectively (George et al., Nature 2015)¹. In that study, the majority of the remaining tumors without mutations in one or both of these genes were attributed to genome rearrangements that were undetectable by exome sequencing but could be observed through WGS. In our own cohort of exome sequenced relapsed SCLCs, TP53 and RB1 were mutated in 97% and 70% of samples, respectively. Similarly, these genes exhibited near-universal loss of heterozygosity (LoH) across the cohort, concordant with the previously reported genomic landscape of primary SCLCs. The LoH data were omitted from our initial submission, and we thank the reviewer for illuminating the need for these clarifying data. We have updated Figures 1 and S3 to include an indication of LoH in evaluated genes and have added the VAF scores associated with the LoH as Supplementary Table S11.

5. The relapsed samples came from several different metastatic sites (I assume that Suppl Table 1, column L that is titled “Tissue Description”) and companion Column I is where the samples came from. Was there any difference in the mutational pattern dependent on where the relapsed biopsy came from?

We are unable to test for a significant difference in observed mutations exome-wide at N=30. Limiting our tests to only those genes reported in Figures 1 and S3, we are able to test for significant differences in mutation rates of each gene in tumors from lymph nodes (N=18) compared to other sites. No significant differences ($q < 0.05$, $p < 0.05$) were found in these comparisons. In evaluating this problem, we determined that other tumor sites were too few in number ($N \leq 5$) to warrant testing. We have added an annotation track to Figures 1 and S3 to clearly indicate the tumor site to aid readers in identifying any possible trends in these data.

6. From the end of the paper I was not clear whether they felt their findings agreed or disagreed with the EZH2/SLFN11 findings. Whatever the result is (that some do and others don't or something else) this needs to be clearly stated.

We did not find recurrent alterations in EZH2 or SLFN11 in samples from patients with relapsed small cell lung cancer. We evaluated SLFN11 expression in our relapsed cohort against the 80 primary samples from George et al. and found no evidence of differential expression for this gene in primary vs. relapsed samples¹. To do this comparison, we compared SLFN11 expression percentiles in each sample type, which yielded the following distribution:

	count	mean	std	min	25%	50%	75%	max
primary	80.0	0.507323	0.283558	0.020202	0.280303	0.510101	0.750000	0.989899
relapse	19.0	0.495481	0.324466	0.010101	0.217172	0.404040	0.767677	1.000000

The median percentile is slightly lower for the relapsed cohort, so we evaluated the distribution of SLFN11 percentile by tumor type, which failed to demonstrate any noticeable clustering of downregulated SLFN11 patients in relapsed SCLC:

While these results do not support the findings of Gardner et al., it is worth noting that these are unpaired primary and relapse samples, and thus we cannot evaluate subtle changes in gene expression between conditions as was performed in that study². We have added the following text to the manuscript to summarize these findings (lines 238-240):

...EZH2-mediated epigenetic reprogramming that led to the silencing of SLFN11 was identified as the mechanism driving induced chemoresistance in this study - a finding which we're unable to reproduce, possibly due to a lack of paired primary-relapse RNA sequencing data.

7. I assume all of the genomics data will be deposited as part of this publication and that, of course is an iron-clad requirement.

In addition to the result datasets provided in the Supplementary tables and methods repository (e.g. fusions, somatic SNVs/indels, CNAs, RNA expression values, ssGSEA enrichment scores), all sequencing data have been uploaded to dbGaP (accession #PHS001049), and this detail is specified at the end of the

manuscript (line 821).

Reviewer #2 (Remarks to the Author):

The authors carried out whole-exome sequencing analyses of relapsed small cell lung cancer (SCLC) to elucidate its mechanisms. They found loss-of-function alterations in the APC gene in relapsed SCLC cases, as well as chemotherapy resistant SCLC cell lines, and underlined involvement of the canonical WNT/ β -catenin signaling activation in relapsed or chemotherapy-resistant SCLCs.

The following points were noticed as issues to be further addressed:

(1) Flow of manuscript. The authors showed whole spectrum of genetic alterations in replaced SCLCs in Figure 1, whereas in the following parts they focused on relatively rare events related to genetic alterations in canonical WNT signaling components and mechanistic analyses to explain how the canonical WNT signaling activation leads to relapse or chemotherapy resistance. I am afraid that the flow of this manuscript was hindered between Figure 1 and the following parts.

We appreciate the reviewer's consideration of the manuscript flow. We agree with this concern and have added section headers to clearly delineate the three primary findings of this manuscript: 1) The similarity in mutational landscape between primary and relapsed SCLC (line 80), 2) the notable difference in neuroendocrine differentiation markers between primary and relapsed SCLC (line 138), and 3) activation of the WNT signaling pathway in the setting of acquired chemoresistance (line 168). We have also revised the text of the manuscript near the beginning of each section to highlight this logical progression. Figure 1B was moved to the supplementary materials as Figure S3, as we agree with the reviewer's assessment that this figure (prior to removing panel b) disrupted the manuscript flow. Finally, we moved some of the text around to better fit this more structured format for the manuscript.

(2) Precise contribution rate of canonical WNT/ β -catenin signaling in SCLC relapse. The authors detected alterations of the APC gene (N507I, Q1204*, H1232Y and E1464fs) in 4 of 30 cases of relapsed SCLCs. Q1204* and E1464fs are loss-of-function APC alterations, whereas functional consequences of N507I and H1232Y APC variations remain unclear. These facts indicate that loss-of-function APC alterations occur in 6.7 % (2 / 30) of relapsed SCLC cases. By contrast, the canonical WNT/ β -catenin signaling cascade is aberrantly activated in tumor cells owing to coding and non-coding alterations in the APC, AXIN1, AXIN2, CTNNB1, RNF43, RSPO2, RSPO3 and ZNRF3 genes as well as canonical WNT ligand elevation in the tumor microenvironment. The precise contribution rate of canonical WNT/ β -catenin signaling activation in SCLC relapse is underestimated in this exome sequencing-based study. The authors are advised to address this issue to improve the value of this report for diverse groups of clinical oncologists.

The authors thank the referee for their insightful comments. We agree with the reviewer that our ability to identify several alterations may have been limited by the fact that these samples were analyzed for genomic alterations through whole-exome sequencing alone. Further analysis has now been performed to examine the extent to which other detectable alterations, such as LoH, affect WNT pathway genes. While the mechanism by which specific mutations impact these pathways remain unclear, the addition of these data clearly illustrate highly recurrent alterations in evaluated WNT signaling genes, which we have added as Figure S3 and further discuss in lines 183-184. In response to the referee's suggestion to highlight this point in the manuscript for the benefit of clinical oncologists, we have added text to the manuscript discussion to relay these findings in lines 221-225. Specifically, we mention:

...WNT signaling genes are likely to be altered through mechanisms other than non-synonymous mutations, which WES alone is unable to detect.

(3) Lack of omics data other than genomics and transcriptomics data. This report mainly depends on genomics and transcriptomics data. Related to (2), the authors need to carry out immunohistochemical analysis of β -catenin to strengthen their claim on the contribution of the canonical WNT/ β -catenin signaling in relapse or chemotherapy-resistance of SCLC.

We agree with the reviewer that examining whole-exome and transcriptome data alone is unlikely to paint a comprehensive picture of the genomic underpinnings that drive chemoresistance. While immunohistochemistry for CTNNB1 was performed on remaining tissues, these data have not been presented in the manuscript, due to several limitations that confound interpretation, as follows.

Previously published studies^{3,4} indicate that almost all SCLC biopsies show positive CTNNB1 staining, therefore only gradations in staining can be compared. In our analysis, of the 5 treatment-naive samples and 23 relapse samples we had available for IHC analysis, no nuclear expression was observed in the treatment-naive samples and was observed in only two of the relapse samples. Furthermore, 40% (n=2) of treatment-naive samples showed membranous staining and 60% (n=3) showed disrupted membranous staining⁴, while only 13% (n=3) of relapse samples showed membranous staining and 78% (n=18) showed disrupted membranous staining. Thus, while we do see differences in CTNNB1 staining by IHC, the small sample size and lack of matched treatment-naive samples for the majority of relapsed samples limited our ability to interpret these findings meaningfully. Additionally, grading intensity of IHC staining is likely to be confounded by factors such as cold ischemic time, duration of fixation, processing, and the subjective nature of interpretation and was therefore not pursued. For these reasons, WNT pathway activity was inferred through ssGSEA analysis of transcriptome data. We have modified the manuscript discussion to acknowledge this limitation of our cohort in lines 243-244:

While these factors limit our ability to perform a number of additional informative analyses (e.g. tumor/relapse paired WNT/ β -catenin immunohistochemistry)...

References

- 1 George, J. *et al.* Comprehensive genomic profiles of small cell lung cancer. *Nature* **524**, 47-53, doi:10.1038/nature14664 (2015).
- 2 Gardner, E. E. *et al.* Chemosensitive Relapse in Small Cell Lung Cancer Proceeds through an EZH2-SLFN11 Axis. *Cancer Cell* **31**, 286-299, doi:10.1016/j.ccell.2017.01.006 (2017).
- 3 Rodríguez-Salas, N. *et al.* Beta-catenin expression pattern in small cell lung cancer: correlation with clinical and evolutive features. *Histol Histopathol* **16**, 353-358 (2001).
- 4 Pelosi, G. *et al.* Alteration of the E-cadherin/beta-catenin cell adhesion system is common in pulmonary neuroendocrine tumors and is an independent predictor of lymph node metastasis in atypical carcinoids. *Cancer* **103**, 1154-1164, doi:10.1002/cncr.20901 (2005).

REVIEWERS' COMMENTS:

Reviewer #1 (Remarks to the Author):

The authors have responded appropriately to all of the reviewers' comments including providing additional clinical information as requested and clarifying the data presentation.

Reviewer #2 (Remarks to the Author):

The authors almost appropriately revised their manuscript.